# INSTRUCTRETRO: INSTRUCTION TUNING POST RETRIEVAL-AUGMENTED PRETRAINING

## ABSTRACT

Pretraining auto-regressive large language models (LLMs) with retrieval demonstrates better perplexity and factual accuracy by leveraging external databases. However, the size of existing pretrained retrieval-augmented LLM is still limited (e.g., *Retro* has 7.5B parameters), which limits the effectiveness of instruction tuning and zero-shot generalization. In this work, we introduce *Retro* 48B, the largest LLM pretrained with retrieval before instruction tuning. Specifically, we continue to pretrain the 43B GPT model on additional 100 billion tokens using the Retro augmentation method by retrieving from 1.2 trillion tokens. The obtained foundation model, Retro 48B, largely outperforms the original 43B GPT in terms of perplexity. After instruction tuning on Retro, *InstructRetro* demonstrates significant improvement over the instruction tuned GPT on zero-shot question answering (QA) tasks. Specifically, the average improvement of InstructRetro is 7% over its GPT counterpart across 8 short-form QA tasks, and 10% over GPT across 4 challenging long-form QA tasks. Surprisingly, we find that one can ablate the encoder from InstructRetro architecture and directly use its decoder backbone, while achieving comparable results. We hypothesize that pretraining with retrieval makes its decoder good at incorporating context for QA. Our results highlights the promising direction to obtain a better GPT decoder for QA through continued pretraining with retrieval before instruction tuning.

## 1 INTRODUCTION

Retrieval helps large language models (LLM) to handle current events, detailed knowledge, proprietary information not in pretraining, and to improve factual grounding (e.g., Nakano et al., 2021; Thoppilan et al., 2022; Borgeaud et al., 2022). In the previous study, pretraining auto-regressive language model with retrieval (i.e., *Retro*) demonstrates successes in reducing perplexity (Borgeaud et al., 2022) and improving factual accuracy (Wang et al., 2023a).

In the past year, the decoder-only auto-regressive LLMs have demonstrated remarkable successes (e.g., OpenAI, 2022; 2023), because *i)* LLMs have been scaled to hundreds of billion parameters (Brown et al., 2020a; Rae et al., 2021; Smith et al., 2022; Chowdhery et al., 2022), *ii)* pretraining corpus has been scaled up to trillions of tokens (Hoffmann et al., 2022; Touvron et al., 2023a;b), and *iii)* instruction tuning (Wei et al., 2022a; Chung et al., 2022) and reinforcement learning from human feedback (RLHF) (Ouyang et al., 2022) recipes have been applied on these pretrained LLMs. In contrast, the pretrained retrieval-augmented language models still have relatively small number of parameters trained limited number of tokens. For example, the auto-regressive *Retro* has 7.5B parameters and is trained on 600B tokens (Borgeaud et al., 2022), *Retro++* has 9.5B parameters and is trained on 330B tokens (Wang et al., 2023a), and T5-based *Atlas* has 11B parameters and is trained with retrieval on maximum 327M tokens (Izacard et al., 2022c). The lack of scaling limits the effectiveness of instruction tuning (Wei et al., 2022a) and other intriguing properties that exist in large language models (Wei et al., 2022b).

In this work, we scale up *Retro* up to 48B parameters, trained on 1.2T tokens in total, i.e., 1.1T tokens for pretraining its GPT backbone, 100B tokens for continued retrieval-augmented pretraining while retrieving from 1.2T tokens. As a result, we can mitigate the zero-shot generalization gap on question answering tasks after applying instruction tuning.

Specifically, we make the following contributions:

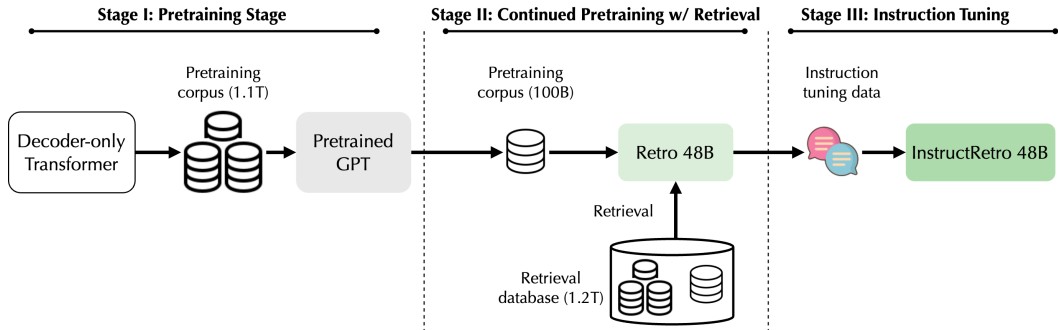

Figure 1: Training pipeline for InstructRetro 48B.

1. We introduce *Retro* 48B, the largest LLM pretrained with retrieval. To save the computation budget, we continue to pretrain a 43B parameter GPT model (originally trained on 1.1T tokens) on adddtional 100B tokens by retrieving from 1.2T tokens. In contrast to *Retro-fitting* (Borgeaud et al., 2022), that freezes pretrained decoder weights, we unfreeze the decoder, jointly train all the parameters and find better perplexity.[1] Notably, we find the perplexity improvement of Retro 48B over its GPT 43B counterpart is still significant even at this scale.

2. After instruction tuning, *InstructRetro* 48B demonstrates strong zero-shot capability to incorporate context for question answering (QA), and significantly outperforms its GPT counterpart using the same instruction tuning recipe. The full pipeline to train InstructRetro is shown in Figure 1.

3. Perhaps surprisingly, we find that one can directly ablate the encoder from *IntructRetro* and still obtain comparable results on zero-shot QA tasks. This highlights the promising direction of obtaining better decoder-only LLMs through continued pretraining with retrieval before instruction tuning.

We organize the rest of the paper as follows. We discuss related work in Section 2, and introduce the continued pretraining of *Retro 48B* in Section 3. We present the instructing tuning in Section 4. We report results in Section 5 and conclude the paper in Section 6.

## 2 RELATED WORK

Retrieval-augmented language models have been established for open domain question answering for years (Karpukhin et al., 2020; Lewis et al., 2020; Guu et al., 2020; Borgeaud et al., 2022; Izacard et al., 2022c). In the previous study, language models have been augmented with retrieval at inference (Khandelwal et al., 2020; Yogatama et al., 2021), fine-tuning (Karpukhin et al., 2020; Lewis et al., 2020; Guu et al., 2020; Huang et al., 2023), and pretraining (Borgeaud et al., 2022; Izacard et al., 2022c; Wang et al., 2023a). Retrieval-augmented pretraining is particularly interesting, as it can largely reduce model perplexity (Borgeaud et al., 2022), enhance factuality (Wang et al., 2023a), and improve downstream task accuracy after task-specific fine-tuning (Izacard et al., 2022c).

In contrast to the state-of-the-art decoder-only LLMs with hundreds of billion parameters (Brown et al., 2020b; Rae et al., 2021; Smith et al., 2022; Chowdhery et al., 2022), the sizes of pretrained retrieval-augmented LLMs are still around 10B parameters (Borgeaud et al., 2022; Wang et al., 2023a; Izacard et al., 2022b), which largely limits the zero-shot generalization capability after instruction tuning (Wei et al., 2022a; Ouyang et al., 2022; Chung et al., 2022). For example, Wei et al. (2022a) finds instruction tuning becomes effective when the decoder-only LLM has around 50B parameters.

Instruction tuning aims to teach LLMs to follow natural language instructions (Wei et al., 2022a; Ouyang et al., 2022; Sanh et al., 2022b; Mishra et al., 2022), which becomes an indispensable ingredient to build the state-of-the-art LLMs for chat and QA tasks (OpenAI, 2022; 2023; Touvron

---

[1]Note that, it turns out that unfreezing of decoder is an important design not only for better perplexity, and it eventually leads to the interesting finding after instruction tuning.

et al., 2023b). In the past years, many high-quality instruction tuning datasets have been created, e.g., FLAN (Chung et al., 2022), OpenAssistant (Köpf et al., 2023), and Dolly (Conover et al., 2023).

## 3 CONTINUED PRETRAINING OF GPT WITH RETRIEVAL

In this section, we start by introducing the preliminaries of *Retro* (Borgeaud et al., 2022) and highlight some key differences between Retro and GPT. We then go through the pretraining details of how we scale up the size of Retro to 48B, a size that has never been studied before.

### 3.1 PRELIMINARIES OF RETRO

Retro (Borgeaud et al., 2022) is an auto-regressive language model pretrained with retrieval augmentation. While Retro shares the backbone of GPT models, Retro differs from GPT by incorporating an additional *Retro encoder*. The Retro encoder is adept at encoding features of retrieved neighbors from *external knowledge bases*. Furthermore, Retro adds *chunk-wise cross-attention* layers within its decoder transformer architecture to integrate retrieved information from the Retro encoder effectively. This design paradigm also makes Retro different from the encoder-decoder architecture (e.g., T5 (Raffel et al., 2020) and Atlas (Izacard et al., 2022b)). The success of scaling decoder-only autoregressive language models (e.g., ChatGPT (OpenAI, 2022) and GPT-4 (OpenAI, 2023)) motivates us to further scale up Retro and understand the potential benifit of retrieval-augmented pretraining.

**Retro encoder** is a shallow bidirectional transformer to encode retrieved neighbors from external databases into dense features. Specifically, in this work, we follow Borgeaud et al. (2022) and use a two-layer bidirectional transformer as the Retro encoder with the same hidden dimension as the Retro backbone decoder. Our preliminary results show that increasing the layers of the Retro encoder does not bring better perplexity on the validation set but increases the computational overhead and model parameters.

**Retrieval database.** Borgeaud et al. (2022) demonstrates that retrieval-augmented pretraining can significantly benefit from large-scale retrieval up to trillions of tokens. To build the retrieval database, we utilize the entire pretraining corpus, but holding out $1\%$ as a validation set. This ensures that both Retro and GPT models are pretrained on an equivalent volume of information from the pretraining corpus. Our retrieval database is a key-value database, where values are chunks of tokens split from the pretraining corpus, and the keys are corresponding BERT embeddings (Devlin et al., 2018). The pretraining corpus consists of 1.2 trillion tokens of English corpus. More details of the pretraining corpus can be found in Appendix §A.1. In summary, our retrieval database comprises 19 billion chunks, with each chunk containing 64 tokens.

**Chunk-wise cross-attention.** Aligning with the chunk-wise design of the retrieval database, Retro splits the input tokens into a sequence of chunks. Specifically, Retro retrieves nearest neighbor chunks using the previous chunk and fuses this information with the context from preceding chunks to guide the generation of the next chunk. Formally, given a input sequence $X$ with $n$ tokens $X = (x_1, ..., x_n)$, Retro splits $X$ into a sequence of $l$ chunks $(C_1, ..., C_l)$ with chunk size $m = \frac{n}{l}$. From a high-level perspective, Retro uses the last $(i-1)$-th chunk $C_{i-1}$ to retrieve $k$ nearest neighbor chunks $\mathcal{N}(C_{i-1})$ from the retrieval database, and fuses the contextual information from the previous chunks $(C_1, ..., C_{i-1})$ and retrieval information from $\mathcal{N}(C_{i-1})$ by cross-attention to guide the generation of the next $(i)$-th chunk $C_i$. To avoid breaking the causality, the autoregressive generation of $i$-th chunk $C_i$ can only use the nearest neighbors of the previous chunk $\mathcal{N}(C_{i-1})$ instead of $\mathcal{N}(C_i)$. In our work, we follow Borgeaud et al. (2022) and retrieve top-$k = 2$ nearest neighbors for each chunk, with chunk size $m = 64$ and the maximum number of tokens $n = 4096$.

### 3.2 RETRO-FITTING: CONTINUED PRETRAINING WITH RETRIEVAL

There are two main challenges of scaling up Retro: the large-scale retrieval database and the huge pretraining cost. To overcome the challenges, we leverage the *Faiss* index (Johnson et al., 2019) to achieve fast approximate nearest neighbor search and *retro-fitting* techniques to reuse the pretrained GPT parameters and save computational cost.

**Retrieval index to the large-scale retrieval database.** We use the Faiss index (Johnson et al., 2019) as the implementation for the dense retriever to search for approximate nearest neighbors in the BERT

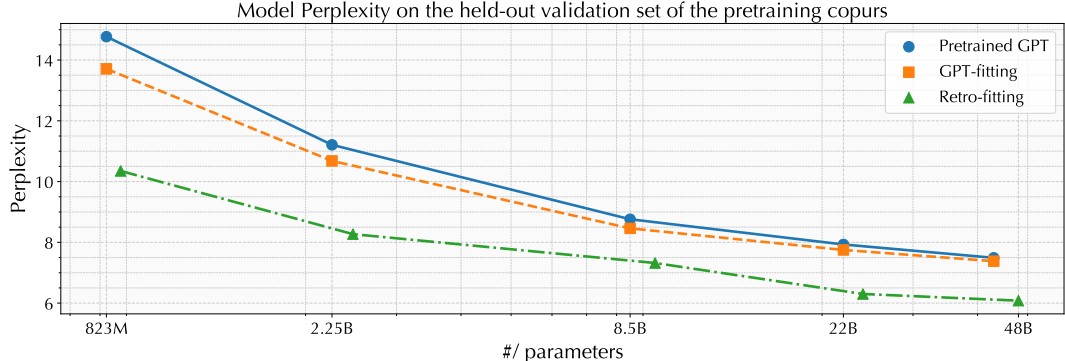

Figure 2: Perplexity evaluation of pretrained GPT models, GPT-fitting, and Retro-fitting models across various parameter sizes on the held-out validation set. In contrast to Borgeaud et al. (2022), we unfreeze all parameters for Retro-fitting. Retro significantly outperforms GPT models, achieving the perplexity comparable to GPT models with 4× larger parameter sizes.

embedding space. We configure the Faiss index to cluster the dense embeddings into $2^{22}$ centroids accelerated with Hierarchical Navigable Small World (HNSW) graphs (Malkov & Yashunin, 2018) to speed up the query. We also encode the embeddings with optimized product quantization (Gray & Neuhoff, 1998; Ge et al., 2014) to compress memory overhead and further improve the query throughput. As a result, we can achieve 4*ms* per query over the whole pretraining corpus averaged for each chunk on a DGX-A100 node. One may find more details in Appendix §B.

**Unfreezing decoder at Retro-fitting.** As Retro shares its backbone decoder with the GPT decoder and only adds around 10% additional parameters for Retro encoder and cross-attention, we can initialize Retro decoder from pretrained GPT models, randomly initialize Retro encoder and cross-attention, and continue pretraining with retrieval, which is named as "*Retro-fitting*". Note that, Borgeaud et al. (2022) freezes the decoder parameters at Retro-fitting. In contrast, we **unfreeze all the decoder parameters** and continue pretraining the entire model. We also conduct an ablation study of Retro-fitting based on a pretraiend GPT of 823M parameters and compare the validation perplexity loss when freezing or unfreezing Retro decoder during pretraining. As shown in Figure 3, given the same training schedules, unfreezing Retro decoder parameters converges faster and demonstrates better validation perplexity, which eventually yields a better Retro decoder to incorporate in-context retrieved evidence, even without a Retro encoder as shown in §5.3. We continue pretraining with retrieval on an additional 100 billion tokens, which is 9% of the pretraining data used for pretrained GPT models. To have a fair comparison, we also continue pretraining GPT foundation models on the same 100 billion tokens, which we name "*GPT-fitting*". More pretraining details can be found in Appendix A.2.

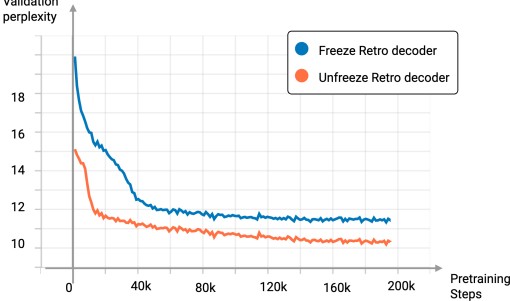

Figure 3: Validation perplexity of Retro-fitting when we freeze or unfreeze Retro decoder during continued pretraining on 100B tokens.

**Base pretrained GPT.** We launch continue pretraining (*i.e.*, GPT-fitting and Retro-fitting) based on pretrained GPT models. Specifically, we pretrain from scratch a set of GPT models with the following parameter sizes: 823M, 2.25B, 8.5B, 22B, and 43B. All of the models are based on

Transformer (Vaswani et al., 2017) with different hidden dimensions, number of layers, and attention heads. We adopt the Sentence Piece tokenizer (Kudo & Richardson, 2018) for both GPT and Retro. We pretrain all models with 1.1 trillion tokens of the pretraining corpus. More details can be found in Appendix §A.

**Perplexity evaluation.** We evaluate the perplexity of GPT foundation models, GPT-fitting models, and Retro-fitting models of varying parameter sizes in Figure 2. The validation corpus consists of 1% held-out samples from the pretraining corpus, which are not used in the pretraining stage, the continued pretraining stage, and the retrieval database to ensure that there is no validation data leakage. From Table 2, one can see that after continued pretraining on additional 100 billion tokens, the perplexity of GPT-fitting slightly improves, while Retro significantly outperforms both GPT and GPT-fitting across different parameter sizes in terms of perplexity. Specifically, Retro achieves even better perplexity than GPT models with $4\times$ larger parameter sizes, and the trend of improvement does not diminish when the parameter sizes of Retro scale up to 48B. We present more evaluation results in §5.3.

## 4    INSTRUCTION TUNING

Instruction tuning can significantly improve the ability of foundation LLMs to follow instructions, thus improving zero-shot results on downstream tasks (e.g., Wei et al., 2022a; Chung et al., 2022). In this section, we further enhance Retro via instruction tuning.

### 4.1    DATASETS BLENDING

Existing instruction tuning methods mainly leverage two training paradigms: supervised fine-tuning on a blend of instruction datasets (Wei et al., 2022a; Chung et al., 2022; Sanh et al., 2022a; Wang et al., 2023b) or reinforcement learning through human feedback (RLHF) (Ouyang et al., 2022; Bai et al., 2022; OpenAI, 2023). Due to the limited open-source human feedback data, we focus on supervised instruction tuning for Retro to unveil the potential of retrieval-augmented LLMs.

We use a blend of high-quality instruction tuning datasets to train LLMs to follow instructions in conversational formats, which include: *i)* a high-quality social dialogue dataset SODA (Kim et al., 2022), *ii)* a long-form QA dataset ELI5 that requires elaborate answers (Fan et al., 2019), *iii)* LLM-generated instructions: Self-Instruct (Wang et al., 2022) and Unnatural Instructions (Honovich et al., 2022), *iv)* FLAN and Chain-of-thought datasets (Chung et al., 2022; Wei et al., 2022c; Longpre et al., 2023), *v)* a private crowd-sourced conversational dataset and public human-written conversation datasets OpenAssistant (Köpf et al., 2023) and Dolly (Conover et al., 2023), and *vi)* samples from the pretraining corpus.

The format of all the instruction tuning data is unified in a conversational way with three roles: "system", "assistant", and "user". The "system" role sets up the tone and style of LLM assistants to give helpful, detailed, and polite answers to the user's questions. The "user" and "assistant" role contains the questions and the corresponding answers from the instruction tuning datasets. We show an example format of the instruction data in Appendix C.1. In total, we collect a total of 128K high-quality samples for instruction tuning.

### 4.2    TRAINING DETAILS

For each training sample, we take the multi-turn conversations between the user and the assistant as context and apply the loss mask only to the last response from the assistant. We use the standard language modeling loss with teacher forcing. Since Wei et al. (2022a) suggests that instruction tuning is most effective with *large* language models, we apply instruction tuning to the GPT-fitting 43B model and the Retro 48B model, naming them "GPT$_{RAG}$-Instruct 43B"[2] and "InstructRetro 48B", respectively. We finetune the LLMs by taking the loss only on the answer part with a batch size of 128 and a learning rate of 5e-6 for 1000 steps with a weight decay of 0.01. We use the Adam optimizer (Kingma & Ba, 2014) with $\beta_1 = 0.9$ and $\beta_2 = 0.98$.

---

[2]We distinguish "GPT$_{RAG}$-Instruct", which uses supervised fine-tuning, from "InstructGPT" (Ouyang et al., 2022), which leverage RLHF for instructing tuning.

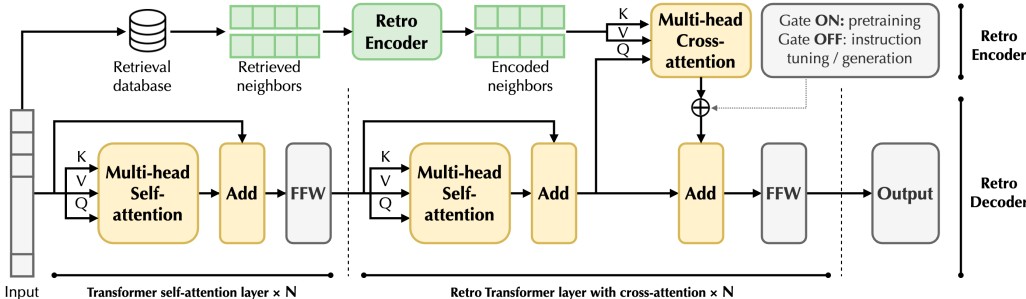

Figure 4: Retro simplified architecture diagram. We omit the layer norm, softmax, and embedding layers for simplicity. We add additional 0/1 gates between cross-attention output and the residual connection from self-attention output. During pretraining, we keep the Retro encoder gate ON (one). During instruction tuning and inference, we turn the Retro encoder gate OFF (zero) when there is no retrieved neighbors.

**Instruction tuning for Retro.** Since the Retro backbone largely shares with GPT models, the training objective of Retro is also the same as GPT models. However, one noticeable difference is that Retro requires retrieval of nearest neighbors for the input instructions, which is not available from all the instruction tuning datasets. Since the instruction tuning data is high-quality, retrieval from the pretraining corpus can yield noisy neighbors, thus not helping improve the model capabilities to follow instructions. We instead skip the cross-attention connection through a manually-set gated mechanism as detailed in Figure 4, which sets the gate to *zero* when retrieved neighbors are not available. During backpropagation, as the cross-attention module and the connected retro encoder are skipped, their parameters are effectively frozen, and only the weights of decoder backbone gets updated. The design makes Retro learn to inference with and without retrieval during instruction tuning, potentially improving the generalization of the Retro decoder. We also leave it as an important future direction to construct retrieval-augmented instruction tuning data for retrieval-augmented generation.

## 5 EXPERIMENT

In this section, we conduct comprehensive studies on the zero-shot capabilities of *InstructRetro* and its GPT counterpart across various downstream tasks to unveil the potential of Retro 48B after instruction tuning.

### 5.1 EXPERIMENTAL SETUP

**Datasets.** We follow the literature of retrieval-augmented generation (RAG) (Lewis et al., 2020; Karpukhin et al., 2020; Izacard et al., 2022a; Wang et al., 2023a) and evaluate InstructRetro 48B and GPT$_{RAG}$-Instruct 43B on a wide range of open-ended Question Answering (QA) datasets. To demonstrate the generalization of instruction tuning, we follow FLAN (Wei et al., 2022a) and primarily focus on *zero-shot evaluation* of QA datasets. Specifically, we consider two categories of open-ended QA datasets: (1) *short-form QA datasets*, which expect short answers (answers within a few tokens), including Natural Question (NQ) (Kwiatkowski et al., 2019), TriviaQA (Joshi et al., 2017), NewsQA (Trischler et al., 2016), SQuAD 1.1 (Rajpurkar et al., 2016), SQuAD 2.0 (Rajpurkar et al., 2018), Quoref (Dasigi et al., 2019), NarrativeQA (Kočiský et al., 2018), DROP (Dua et al., 2019); (2) *long-form QA datasets*, which expect longer answer spans within a few sentences. As the ELI5 (Fan et al., 2019) dataset is used in the instruction tuning stage, we exclude it from the zero-shot long-form QA evaluation. We instead focus on the following long-form QA datasets: doc2dial (Feng et al., 2020), two proprietary annotated car manual datasets (people ask questions about the particular model of a car), and another proprietary annotated IT dataset. We note that we apply retrieval-augmented generation (RAG) for InstructRetro and GPT$_{RAG}$-Instruct by using the task-provided context of QA tasks (*e.g.*, SQuAD 1.1 and 2.0) or state-of-the-art retrievers to retrieve high-quality contexts from the task-specific corpus (*e.g.*, DPR (Karpukhin et al., 2020) for NQ and TriviaQA and DRAGON+ (Lin et al., 2023) for doc2dial and other long-form QA datasets).

Table 1: Zero-shot evaluation on eight short-form QA datasets. The average *relative* improvement of InstructRetro across the short-form QA tasks is 7% over GPT$_{RAG}$-Instruct.

| Task | NQ | TriviaQA | NewsQA | SQuAD 2.0 | SQuAD 1.1 | Quoref | NarrativeQA | DROP |
|---|---|---|---|---|---|---|---|---|
| Metric | EM | EM | F1 | F1 / EM | F1 / EM | F1 | F1 | F1 |
| *Without Retrieval-Augmented Generation (RAG)* | | | | | | | | |
| GPT-3 175B (Brown et al.) (Chowdhery et al.) | 14.6 | 64.3 | - | 59.5 / 52.6 | - | - | - | 23.6 |
| PaLM 2 -L (Chowdhery et al.) | 37.5 | - | - | - / - | - | - | - | - |
| Llama 65B (Touvron et al.) | 23.8 | 68.2 | - | - / - | - / 79.4 | - | - | - |
| Llama 2 70B (Touvron et al.) | 25.3 | - | - | - / - | - / **80.7** | - | - | - |
| GLaM 64B (Du et al.) | 24.7 | **71.3** | - | 71.1 / 64.7 | - / - | - | - | **57.3** |
| FLAN-LaMDA 137B (Wei et al.) | 20.7 | 68.1 | - | 44.2 / - | 80.1 / - | - | - | 22.7 |
| *With Retrieval-Augmented Generation (RAG)* | | | | | | | | |
| Retro 7.5B (Borgeaud et al.) | 8.9 | 36.0 | - | - / - | - | - | - | - |
| Retro++ 9B (Wang et al.) | 25.8 | 48.3 | - | - / - | - | - | - | - |
| Atlas 11B (Izacard et al.) | 26.7 | 56.9 | - | - / - | - / - | - | - | - |
| Raven 11B (Huang et al.) | 29.6 | 65.7 | - | - / - | - / - | - | - | - |
| GPT$_{RAG}$-Instruct 43B | 37.0 | 65.0 | 52.4 | 70.7 / 64.3 | 72.4 / 65.8 | 71.5 | 53.9 | 51.8 |
| InstructRetro 48B (Avg: +7%) | **38.9** (+5.14%) | 65.6 (+0.92%) | **57.4** (+9.54%) | **75.6 / 69.3** (+6.93%) | 77.1 / 70.4 (+6.49%) | **76.2** (+6.57%) | **60.0** (+11.32%) | 54.8 (+5.79%) |

**Baselines.** GPT$_{RAG}$-Instruct 43B is our main baseline as both GPT$_{RAG}$-Instruct 43B and InstructRetro 48B have the same GPT decoder hyper-parameters (e.g., number of transformer layers, hidden sizes, etc.) and was pretrained and instruction-tuned on the same amount of data. In addition, we also compare a wide range of state-of-the-art large language models with comparable or larger sizes, including GPT-3 175B (Brown et al., 2020b), Llama 65B (Touvron et al., 2023a), Llama 2 70B (Touvron et al., 2023b), GLaM 64B (Du et al., 2021), and FLAN-LaMDA 137B (Wei et al., 2022a). Furthermore, we also compare InstructRetro 48B with existing retrieval-augmented LLMs, including Retro 7.5B (Borgeaud et al., 2022), Retro++ 9B (Wang et al., 2023a), Atlas 11B (Izacard et al., 2022b), and Raven 11B (Huang et al., 2023).

**Implementation details.** We use greedy decoding for open-ended QA with the max output length to be 256. We truncate the generation when we encounter the special token |<end-of-document>| or role-switching from "Assistant" to "User" when completing the conversation. All of the QA tasks are re-formated in the conversational format. An example from the SQuAD 1.1 dataset in the conversational prompt format is shown in Appendix Table 11. Most of the QA tasks provide retrieved relevant contexts, which are also incorporated into the prompt. As the Natural Question and TriviaQA datasets do not provide relevant contexts, we take the top-5 DPR-retrieved passages (Karpukhin et al., 2020) as contexts and put them into the prompts. As we do instruction tuning on Retro without enabling its encoder, we also bypass the Retro encoder during evaluation, **making it serve solely as a GPT decoder** to align with the instruction tuning behaviors.

Table 2: Zero-shot evaluation on four long-form QA datasets. We use F1 as the evaluation metric. The average *relative* improvement of InstructRetro across the long-form QA tasks is 10% over GPT_{RAG}-Instruct.

|  | doc2dial | Car Manual Doc #1 | Car Manual Doc #2 | IT Doc |
|---|---|---|---|---|
| GPT_{RAG}-Instruct 43B | 32.87 | 58.18 | 50.88 | 31.40 |
| InstructRetro 48B | **35.74** | **63.52** | **57.49** | **34.08** |
| (avg: +10%) | (+8.73%) | (+9.18%) | (+12.99%) | (+8.54%) |

## 5.2 ZERO-SHOT EVALUATION AFTER INSTRUCTION TUNING

We present the zero-shot evaluation results across eight short-form QA datasets in Table 1. We also apply InstructRetro to four open-ended long-form QA datasets, as detailed in Table 2. These datasets are representative of real-world applications, including chatbots for IT support and customer service.

**Instruction tuning post retrieval-augmented pretraining yields a better GPT decoder.** From both Table 1 and 2, we observe that InstructRetro shows consistent accuracy improvement upon its counterpart GPT_{RAG}-Instruct across different datasets for both short-form and long-form QA. Notably, the average relative improvement of InstructRetro across all the short-form datasets is around 7%. Note that, InstructRetro bypasses its Retro encoder during evaluation, operating solely as a GPT decoder. Given that both InstructRetro and GPT_{RAG}-Instruct are pretrained and instruction tuned with identical datasets, hyper-parameters, and evaluation prompts, we attribute this consistent improvement to the training recipe of InstructRetro, which leverages continued pretraining with retrieval before instruction tuning. To have a deeper understanding, we execute an ablation study, detailed in §5.3.

From Table 1, we also show that our InstructRetro provides compelling performance than other state-of-the-art LLMs. For example, InstructRetro 48B achieves better accuracy than GLaM 64B on multiple datasets (e.g., NQ and SQuAD 2.0), close to FLAN-LaMDA 137B.

**InstructRetro demonstrates larger improvement on long-form QA datasets.** When comparing the results of InstructRetro on short-form QA datasets and long-form QA datasets, we observe InstructRetro demonstrates large relative accuracy improvements, achieving 10% over the GPT_{RAG}-Instruct. As long-form QA tasks are generally more challenging than short-form QA tasks, such improvements further demonstrate the potential of retrieval-augmented pretraining.

## 5.3 ABLATION STUIDES

In this section, we conduct ablation studies to understand the source of improvements for InstructRetro. We show that both retrieval-augmented pretraining and instruction tuning are indispensable to unlock the potential of retrieval-augmented LLMs.

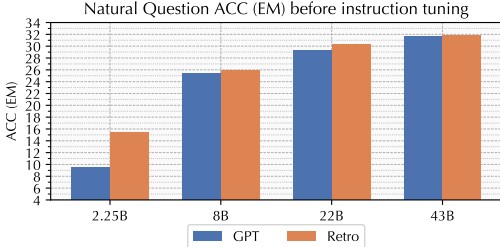

(a) Before instruction tuning, the improvement of retrieval augmentation saturates when the size scales up.

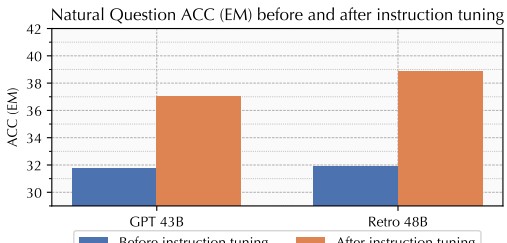

(b) Instruction tuning further unveils the potential of retrieval augmentation even when the size scales up.

Figure 5: Zero-shot accuracy (EM) of GPT and Retro before and after instruction tuning evaluated on the Natural Question dataset.

**Ablation study on instruction tuning.** To understand how instruction tuning improves retrieval-augmented pretraining, we show the zero-shot accuracy (Exact Match) of Retro and GPT on the Natural Question dataset before and after instruction tuning, as detailed in Figure 5. We observe that

Table 3: Zero-shot evaluation on short-form QA tasks. We ablate with encoder (enc.) and without it.

| Task | NQ | TriviaQA | NewsQA | SQuAD 2.0 | SQuAD 1.1 | Quoref | NarrativeQA | DROP |
|---|---|---|---|---|---|---|---|---|
| Metric | EM | EM | F1 | F1 / EM | F1 / Em | F1 | F1 | F1 |
| InstructRetro w/ enc. | 38.6 | 65.4 | 57.0 | 74.8 / 67.7 | 76.4 / 69.0 | 76.1 | 59.8 | 54.6 |
| InstructRetro w/o enc. | **38.9** | **65.6** | **57.4** | **75.6 / 69.3** | **77.1 / 70.4** | **76.2** | **60.0** | **54.8** |

Table 4: Zero-shot evaluation on long-form QA tasks. We use F1 as the evaluation metric. We ablate with encoder (enc.) and without it.

| | doc2dial | Car Manual Doc 1 | Car Manual Doc 2 | IT Doc |
|---|---|---|---|---|
| InstructRetro w/ enc. | **35.95** | 63.16 | 56.82 | **34.07** |
| InstructRetro w/o enc. | 35.74 | **63.52** | **57.49** | 33.71 |

Retro achieves significantly better zero-shot accuracy than GPT when the number of parameters is relatively small (e.g., 2.25B). However, when scaling the size of parameters, the zero-shot performances of both GPT and Retro start to saturate. We hypothesize that this saturation is mainly due to the poor instruction following abilities of both pretrained foundation GPT and Retro models.

To improve their instruction following ability, we apply instruction tuning to further fine-tune both Retro 48B and GPT 43B. Instruction tuning largely mitigate the instruction following bottleneck for both GPT and Retro, resulting in a significant increase of their zero-shot performance on downstream tasks, respectively. Furthermore, once this bottleneck is alleviated, the benefits of retrieval augmentation at pretraining become more pronounced, as InstructRetro excels in leveraging and integrating evidence from retrieved context. Thus, we observe significant improvement of InstructRetro over GPT$_{RAG}$-Instruct again in Figure 5b. The same trend also holds for the TriviaQA dataset, as shown in Appendix Figure 6. This ablation study confirms that our training recipe, both retrieval-augmented pretraining and instruction tuning are important for achieving high performance in QA tasks.

**Ablation study on Retro encoder.** We enable the Retro encoder for retrieval-augmented pretraining, while disabling the Retro encoder for instruction tuning due to the lack of retrieved high-quality neighbors. To understand whether enabling the Retro encoder can further help the evaluation, we further conduct an ablation study and compare the zero-shot accuracy with or without the Retro encoder in Table 3 and 4. When enabling the Retro encoder, we put the top-2 neighbors in the encoder to align with the pretraining behavior.

From Table 3 and 4, the accuracy gap between the two variant is marginal. Disabling the Retro encoder yields very slightly better performance on average than when it is active. This suggests that although Retro is proficiently trained to infer both with and without the neighbors in the encoder, it is more optimal to align with the instruction tuning protocols and bypass the Retro encoder during evaluation. We think it is an important and promising future research direction to explore retrieval-augmented instruction tuning with the Retro encoder activated, especially when high-quality retrieval-augmented instruction data is available. We also conduct additional ablation studies on the impact of retrieval-augmentation generation (RAG) in Appendix §D.2 and include additional experimental results on the summarization tasks in Appendix §D.3.

## 6 CONCLUSION

In this paper, we introduce InstructRetro 48B, the largest LLM with retrieval-augmented pretraining and instruction tuning. Specifically, we start from a pretrained GPT model, and continue pretrain the mddel with retrieval, which yields the retrieval-augmented foundation model Retro 48B. After applying instruction tuning to Retro, InstructRetro 48B unveils the potential of retrieval-augmented pretraining and demonstrates significant zero-shot accuracy improvement over its GPT counterpart through our extensive experiments on a wide range of open-ended QA tasks. Moreover, our novel findings show that only using the GPT decoder backbone of InstructRetro can achieve the very comparable accuracy, which sheds light on a promising direction to obtain a better GPT decoder for QA through retrieval-augmented pretraining before instruction tuning.

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

## A  PRETRAINING DETAILS

### A.1  PRETRAINING CORPUS

We prepared a pretraining dataset consisting of around 1.2 trillion tokens from English natural language data. Specifically, it consists of web-crawl data from Common Crawl, news data, conversational data, book data (e.g., Book3 and Book-Corpus2 from the Pile dataset (Gao et al., 2020)), scientific and multi-domain data (e.g., Wikipedia and the BigScience ROOTS corpus (Laurençon et al., 2022)).

### A.2  CONTINUED PRETRAINING SCHEDULES

Based on pretrained GPT models, we further pretrain Retro with retrieval augmentation on additional 100 billion tokens, which is around 25M samples with sequence length set to 4096. We list the pretraining hyper-parameter details of Retro-fitting in Table 5. GPT-fitting uses the same training schedules as Retro-fitting.

All models use Adam optimizer (Kingma & Ba, 2014) with $\beta_1 = 0.9$ and $\beta_2 = 0.95$. We employ the learning rate (LR) decay schedules with LR warmup samples of 16667 and LR decay samples of 23750000.

Table 5: Detailed pretraining setup for standard pre-trained LMs and InstructRetro.

| Models Size | LR | min LR | LR Decay Styles | Batch Size | Pretraining Steps |
|---|---|---|---|---|---|
| 823M | 2e-5 | 2e-6 | cosine | 128 | 195.2k |
| 2.25B | 2e-5 | 2e-6 | cosine | 256 | 97.6k |
| 8.5B | 1e-5 | 1e-6 | cosine | 512 | 48.8K |
| 22B | 1e-5 | 1e-6 | cosine | 512 | 48.8K |
| 43B | 9e-6 | 9e-7 | cosine | 768 | 32.5k |

### A.3  COMPUTATIONAL COST FOR CONTINUED PRETRAINING

We present the detailed computational cost of the continued pretraining step on additional 100B tokens for both Retro and GPT across different sizes in Table 6. We can see that pretraining Retro brings around additional 35% computational overhead than pretraining GPT, which mainly comes from the Retro encoder and cross-chunk attention to incorporate and fuse the retrieved neighbor information. Moreover, we can see that scaling up the size of Retro does not bring more computational overhead and remains around 35%, shedding light on a promising way to retrieval-augmented pretraining.

Table 6: Pretraining cost of the continued pretraining on 100B tokens for Retro and GPT across different sizes.

|  | GPT on 100B token | Retro on 100B token | Additional Overhead |
|---|---|---|---|
| 800M | 1408 GPU Hours | 1920 GPU Hours | 36% |
| 2B | 3226 GPU Hours | 4096 GPU Hours | 27% |
| 8B | 12698 GPU Hours | 17325 GPU Hours | 37% |
| 22B | 37888 GPU Hours | 52152 GPU Hours | 37% |
| 43B | 53329 GPU Hours | 69995 GPU Hours | 31% |

## B  DETAILS OF RETRIEVAL DATABASE

**Retrieval Database.**  We use the whole pretraining corpus as our retrieval database, consisting of 1.2 trillion tokens as mentioned in Appendix §A.1. Our pretraining dataset with 1.2 trillion tokens yields a retrieval database consisting of 19B chunks in total with chunk size $m = 64$. To support fast similarity searches with billions of chunks, we implement the database index with Faiss index (Johnson et al., 2019). Given the BERT embeddings of an input chunk $C_i$, Faiss can return the approximate $k$ nearest neighbor of $C_i$ within a few milliseconds.

### B.1 FAISS INDEX CONFIGURATION

We use the Faiss index (Johnson et al., 2019) as the implementation for the dense retriever to search for approximate nearest neighbors in the BERT embedding space. We configure the Faiss index as follows:

- **Preprocessing**: We use Optimized Product Quantization (Ge et al., 2014) to apply a rotation to the input vectors to make them more amenable to PQ coding (Gray & Neuhoff, 1998).
- **Indexer**: We use Inverted File Index (IVF) with $2^{22}$ centroids and accelerate it with Hierarchical Navigable Small World (HNSW) graphs (Malkov & Yashunin, 2018).
- **Encoding**: We adopt PQ encoding that compresses the dense embedding vector into 64 bits.

As a result, we can achieve *4ms* per query over the whole pretraining corpus via batch queries averaged for each chunk with less than 1TB memory usage as our max throughput. Given a single query, the latency of the response is around $0.1s$ per query. We also note that increasing the number of $K$ in the query does not yield slower query speed. During pretraining, we follow Borgeaud et al. (2022) to pre-compute the nearest neighbors and save the data for pretraining.

### B.2 COMPUTATIONAL COST ON BUILDING RETRIEVAL DATABASE

Building a Faiss index involves several steps. We detail each step with its associated computational cost as below:

- **Embedding the retrieval database into dense BERT embeddings.** Given the chunk size of $m = 64$ tokens, we embed every chunk of text corpus with BERT-large-cased. The computational cost to embed the text corpus is around 6.22M chunks per GPU hour given one A100 GPU. For our 19B chunk database, it takes around 3054 GPU hours in total.
- **Train the Faiss index.** This involves determining a smaller number of centroids to cluster the whole corpus embeddings and initializing the HNSW graph. The computational cost of training the Faiss index depends on the number of corpus embeddings and the number of centroids. Given our setup, we train the faiss index based on 600M chunks uniformly sampled from the whole retrieval database. The computational cost of this step is less than 4 hours with one DGX A100 node.
- **Add the embedded corpus to the Faiss index.** After the index has been trained, the index centroids and HNSW graph are determined, but the index itself is still empty. In this step, we add the whole dense corpus embeddings to the index data structure. The computational cost of adding the corpus to the index is around 192 CPU hours within one DGX A100 node. Moreover, it can be purely done within a CPU node to save computational cost.
- **Query the Faiss index.** As mentioned above, we can achieve *4ms* per query over the whole pretraining corpus via batch queries averaged for each chunk with less than 1TB memory usage as our max throughput. The computational cost to query over 100B tokens in our continued pretraining step is around 1736 CPU hours within a DGX A100 node. Moreover, this step can also be purely done within a CPU node to save computational cost and can run in parallel to further speed up the querying.

In summary, the overall computational cost of building Faiss index is marginal compared to the pretraining cost, especially considering the benefits of retrieval-augmentation pretraining, which further unlocks the potential of instruction tuning. Thus we believe that it is a promising direction to pretrain with retrieval augmentation.

### B.3 ABLATION STUDIES ON FAISS INDEX CONFIRATIONS

**Faiss training-time configuration.** We conduct ablation studies on the quantization techniques using two index configurations on two datasets: the whole pretraining dataset and the Wikipedia Corpus. We highlight the configuration setup in Table 7 below.

Following the official guide of Faiss[3], we initialize two Faiss indexes based on the sizes of two retrieval databases: the full pretraining corpus with 19B chunks and the Wikipedia corpus with 66M chunks. We applied product quantization (Ge et al., 2014; Gray & Neuhoff, 1998) to the full

---

[3]https://github.com/facebookresearch/faiss/wiki/Guidelines-to-choose-an-index

Table 7: Ablation studies on Faiss product quantization (PQ) on two different retrieval databases.

| | | Retrieval Index for Full Pretraining Corpus | Retrieval Index for Wikipedia Corpus |
|---|---|---|---|
| #/ chunks | | 19B | 66M |
| Configuration | Dimension Reduction Approximate Search Encoding | OPQ64_128 IVF4194304_HNSW32 PQ64 | No Reduction IVF262144_HNSW32 Flat Encoding |
| Query Speed | K=2 K=20 K=200 K=2000 | 0.004 s/query 0.004 s/query 0.0045 s/query 0.004 s/query | 0.01 s/query 0.01 s/query 0.01 s/query 0.01 s/query |

pretraining corpus to reduce the dimensionality and save the index memory to support loading the full pretraining corpus, while applying uncompressed flat encoding to the Wikipedia corpus as a comparison. We benchmark the querying speed for a batch of 40K dense embeddings and evaluate the query speed for two indexes.

From Table 7, we can see that applying product quantization can not only help compress the index and save memory usage but also help improve the query speed, which is critical when scaling up the retrieval database. We can also see that increasing the number of $K$ for $K$ nearest neighbor searchers barely impacts the query speed.

**Faiss query-time configuration.** For our index configuration with interveted file index structures and HNSW graph, the hyper-parameter `nprobe` and `efSearch` play important roles in the query time of Faiss, as detailed in Table 8.

Table 8: Important querying-time hyper-parameters for our Faiss index.

| index type | Index class | runtime parameter | comments |
|---|---|---|---|
| IVF*, IMI2x* | IndexIVF* | nprobe | the main parameter to adjust the speed-precision tradeoff |
| HNSW* | IndexHNSW | efSearch | the depth of the HNSW search |

To select a proper set of query-time hyper-parameters with a good tradeoff of recall and speed, we conduct ablation studies with varying `nprobe` and `efSearch`. Specifically, we use the retrieval index built on the whole pretraining corpus, query the index with randomly sampled 10K chunks from the pretraining corpus, and evaluate the recall accuracy of retrieving the query chunk itself given top-$K = 2000$. The query time and corresponding recall accuracy with different hyper-parameters are shown in Table 9. We thus select `nprobe`=4096 and `efSearch`=32 in our retrieval-augmented pretraining, considering its fast querying time without sacrificing the recall accuracy significantly.

Table 9: Trade-off between speed and recall accuracy given different query-time hyper-parameters.

| | Time (per query) | Acc (K=2000) |
|---|---|---|
| efSearch=256, nprobe=65536 | 21ms | 1.00 |
| efSearch=64, nprobe=16384 | 6ms | 0.96 |
| **efSearch=32, nprobe=4096** | **4ms** | **0.93** |
| efSearch=16, nprobe=4096 | 2.5ms | 0.88 |

## C    QUALITATIVE EXAMPLES

### C.1    AN EXAMPLE FROM THE INSTRUCTION TUNING DATA

Table 10 below is a data example from the instruction tuning data.

Table 10: A sample conversation from our instruction tuning dataset. The black part serves as the context, while the blue part is masked as the answer mask, on which the LM loss is computed.

| Speaker | Conversation |
|---|---|
| System | This is a chat between a user and an artificial intelligence assistant. The assistant gives helpful, detailed, and polite answers to the user's questions based on the context. The assistant should also indicate when the answer cannot be found in the context. |
| User | My bicycle tire has a puncture and while I have a pump with me, I don't have a patch repair kit. Is there a way to keep the tire from running out of air until I can get to a bicycle shop? |
| Assistant | Yes, you could use duct tape to seal the puncture. |
| User | Can you give me a tutorial on how to use the duct tape to seal the puncture? |
| Assistant | Sure, here's a tutorial on how to use duct tape to seal a puncture in a tire:
1. Start by cleaning the area around the puncture with a damp cloth.
2. Cut a piece of duct tape large enough to cover the puncture, plus a few millimeters of extra.
3. Remove the backing from the duct tape, and press it firmly over the puncture.
4. Rub the tape firmly with your fingers to ensure a good seal.
5. You can also use a roller or a flat item like a ruler to make sure that the tape is pressed firmly against the puncture.
6. Trim away any excess tape, and your puncture should now be sealed. |

## C.2   AN EXAMPLE FROM THE DOWNSTREAM QA DATASET: SQUAD 1.1

We re-format all of the QA tasks in the conversational format. An example from the SQuAD 1.1 dataset with the conversational format is shown in the Table 11 below.

Table 11: A sample conversation prompt template from the SQuAD 1.1 dataset.

| Speaker | Conversation |
|---|---|
| System | System: This is a chat between a user and an artificial intelligence assistant. The assistant gives helpful, detailed, and polite answers to the user's questions based on the context. The assistant should also indicate when the answer cannot be found in the context. |
| User | title: , source: The pound-force has a metric counterpart, less commonly used than the newton: the kilogram-force (kgf) (sometimes kilopond), is the force exerted by standard gravity on one kilogram of mass. The kilogram-force leads to an alternate, but rarely used unit of mass: the metric slug (sometimes mug or hyl) is that mass that accelerates at 1 ms-2 when subjected to a force of 1 kgf. The kilogram-force is not a part of the modern SI system, and is generally deprecated; however it still sees use for some purposes as expressing aircraft weight, jet thrust, bicycle spoke tension, torque wrench settings and engine output torque. Other arcane units of force include the sthène, which is equivalent to 1000 N, and the kip, which is equivalent to 1000 lbf.

Based on the above article, answer a question. What is the seldom used force unit equal to one thousand newtons? |
| Assistant | The answer is |

# D  ADDITIONAL EXPERIMENTAL RESULTS

## D.1  DOWNSTREAM TASK ACCURACY OF PRETRAINED MODELS BEFORE INSTRUCTION TUNING

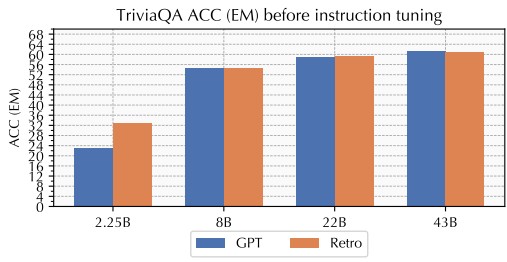

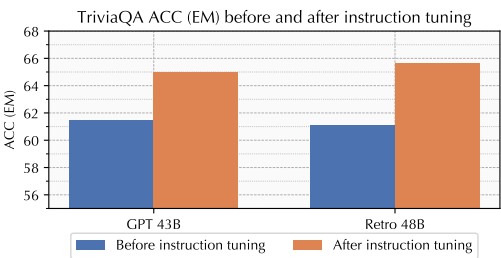

(a) Before instruction tuning, the improvement of retrieval augmentation saturates when the size scales up.

(b) Instruction tuning further unveils the potential of retrieval augmentation even when the size scales up.

Figure 6: Zero-shot accuracy (EM) of GPT and Retro before and after instruction tuning evaluated on the TriviaQA dataset.

## D.2  ABLATION STUDIES ON RETRIEVAL AUGMENTATION

We apply retrieval-augmented generation (RAG) across our main experiments. In this subsection, we conduct ablation studies on the impact of RAG on open-domain QA datasets Natural Question (NQ) and TriviaQA for both GPT-Instruct and InstructRetro. We set the Retro encoder gate OFF in the ablation studies. The results are shown in Table 12 and 13.

Table 12: Accuracy gap between w/ and w/o retrieval-augmented generation (RAG) on NQ

| EM Score | GPT-Instruct | InstructRetro |
|----------|--------------|---------------|
| w/o RAG  | 21.2         | **21.8**      |
| w/ RAG   | 37.0         | **38.9**      |

Table 13: Accuracy gap between w/ and w/o retrieval-augmented generation (RAG) on TriviaQA

| EM Score | GPT-Instruct | InstructRetro |
|----------|--------------|---------------|
| w/o RAG  | 53.6         | **54.5**      |
| w/ RAG   | 65.0         | **65.6**      |

From Table 12 and 13, we can see that the accuracy gap between with or without RAG is significant. For example, the EM scores of NQ surges from 21.8 to 38.9 for InstructRetro. The improvement margin is significant GPT-Instruct as well. Moreover, we observe that InstructRetro consistently outperforms GPT-Instruct across different tasks, even without retrieval augmentation. This further confirms that instruction tuning after retrieval-augmented pretraining can help yield a better GPT decoder.

## D.3  EXPRIMENTAL RESULTS ON SUMMARIZATION TASKS

To demonstrate the generalizability of InstructRetro, we extend our experiments from QA tasks to summarization tasks, focusing on three summarization datasets: QMSum (Zhong et al., 2021), SummScreenFD (Chen et al., 2021), and GovReport (Huang et al., 2021). Following the official metrics, we report the geometric mean of ROUGE scores (*i.e.*, ROUGE1/2/L) for these summarization tasks. The zero-shot evaluation results are shown in the table below.

| ROUGE scores | GovReport | SummScreenFD | QMSum |
|--------------|-----------|--------------|-------|
| GPT$_{RAG}$-Instruct | 12.59 | 10.43 | 15.06 |
| InstructRetro | **17.46** | **10.93** | **15.61** |

From the table above, we observe that InstructRetro consistently outperforms the GPT-Instruct on these summarization tasks, especially on the GovReport dataset with 4.87 ROUGE score improvement.

This experiment further confirms the generalizability of IntructRetro after instruction tuning and indicates that Instruction tuning post retrieval-augmented pretraining yields a better GPT decoder.

