# OpenReview forum: "InstructRetro: Instruction Tuning post Retrieval-Augmented Pretraining"
_ICLR.cc/2024/Conference — Submitted to ICLR 2024_

### Official Review · Reviewer_gdXh · 2023-11-01

**Soundness:** 3 good
**Presentation:** 2 fair
**Contribution:** 3 good
**Rating:** 8
**Confidence:** 4

**Summary:**

This work introduces (1) RETRO-48B, the largest LM pre-trained with retrieval using a 1.2T token database following the RETRO (Borgeaud et al., 2022) architecture, and (2) instruction-tuning RETRO-48B on diverse instruction-response pairs. To achieve (1), they explore multiple quantization and efficient indexing methods to speed up retrieval from 1.2T token storage and unfreeze decoder parameters during retrofitting. Experimental results show that RETRO-48B constantly achieves better perplexity than their continued pre-trained GPT-48B (GPT-fitting) or pre-trained GPT. For (2), they skip the cross attention through a manually set gate mechanism when retrieved neighbors are not available and conduct instruction-tuning on diverse instruction-tuning datasets (e.g., Open Assistant). Their experimental results show that InstructRetro even without an encoder (retrieved chunks) can significantly outperform GPT-48B.

I am impressed by the first contribution of this work -- making retrieval from 1.2T under 4ms only using a DGX-a100 node is difficult--RETRO has not yet been open-sourced and no one has reproduced their results at 1+T scale yet. It is also exciting to see that pre-training with retrieval can even enhance larger base LM (43B) as in the community there isn’t a clear consensus on whether retrieval still helps on a larger scale or whether larger LMs already encode necessary information without retrieval. Especially if the authors can open-source the code and checkpoints, it can inspire follow-up research on scaling retrieval-augmented LM pre-training. I wish they provided ablations on different quantization methods to see whether different techniques can affect final performance or not, as conducting ablations on invidious quantization techniques can also be really expensive. Also given that the great progress of pre-training retrieval models, I wonder if we could replace a frozen BERT-base retriever with a more recent and competitive retrieval model e.g., Contriever.

On the other hand, I am not fully convinced by the contribution (2). The descriptions of the instruction-tuning parts are confusing, and making it difficult to understand the concrete setup. Is retrieval or encoder indeed used instruction-tuning, or is skipped most of the time? Figure 1 seems to indicate that retrieval is skipped during instruction tuning while the text in Section 3.2. says retrieval is skipped only when retrieval context is not available. But there are no clear descriptions of what "retrieval context is not available". I listed detailed questions in Weaknesses and Questions.
If it’s mostly a standard instruction tuning without retrieval on top of RETRO-48B, I don’t think it’s that technically novel. Also, the experimental results indicate that removing the encoder block doesn’t affect the performance that much, which contradicts prior findings where retrieving documents gives large improvements on top of strong LLMs in tasks like OpenQA. While the authors argue this indicates pre-training with retrieval can enhance a decoder-only LM, this result makes me question the effectiveness of instruction tuning or the proposed model's ability to use retrieved context. I suspect the limited deterioration may be from the limited performance of the retrieval component itself (i.e., the BERT encoder is far from the current SOTA retrieval system in the same parameter scale) but it is unclear as there are not many ablations or analyses on different decision choices.

Overall, I think (1) provides strong technical contributions, but many questions are left in the second part, in terms of implementations and results. I suspect more clarification,  improvement of presentations or analysis might help, and am happy to increase my score once I am convinced.

**Strengths:**

- This paper introduces the largest scale of LM pre-trained with retrieval (RETRO-48B).
- They retrieve relevant chunks from a 1.2T token datastore, and by extensive quantization and efficiency techniques they make retrieval fast and scalable.
- They further instruction-tune RETRO-48B on diverse instruction-response pairs.

**Weaknesses:**

I like this paper and believe this paper provides great technical contributions in terms of pre-training retrieval-augmented LM at scale. On the other hand, I have several concerns, especially for the instruction tuning part and their downstream task evaluations. That being said, my concerns partially come from confusion between inconsistent descriptions in the paper, and I am happy to increase my score once I am convinced during the discussion period.


**1. Technical novelty**

Introducing RETRO-like architecture on top of a decoder-only model has been recently studied by [Wang et al. (2023)](https://arxiv.org/abs/2304.06762). Although RETRO-48B is the order of magnitude larger in terms of parameter counts and index size and they employed a lot of techniques to achieve this (Section B), I wish they had provided a more detailed analysis of individual quantization techniques to help the community learn more from this work. Unfreezing decoder parameters is one notable difference from RETRO, but I am not sure if this provides sufficient contributions. The instruction-tuning results seem to be rather mixed and some details aren't super clear to me (discussed below) if it's a standard instruction-tuning, I don't think it provides sufficient contribution either.

**2. Unclear details of instruction-tuning**

The biggest question I had about the instruction-tuning part is in how much of the instruction-tuning, retrieval is indeed used during the instruction-tuning stage.

> However, one noticeable difference is that Retro requires retrieval of nearest neighbors for the input instructions, which is not available from all the instruction tuning datasets. Since the instruction tuning data is high-quality, retrieval from the pertaining corpus can yield noisy neighbors, thus not helping improve the model capabilities to follow instructions. We instead skip the cross-attention connection through a manually-set gated mechanism, which sets the gate to zero when retrieved neighbors are not available.

The description says that the gate is set to zero when retrieved context is not available, but it's unclear what retrieved-context is unavailable means. None of the instruction-following datasets such as OpenAssistant comes with a pre-given retrieved document. Does this mean retrieval is always off during pre-training? Or it is used for some datasets with pre-given context? The lack of details makes it difficult what's new for the instruction-tuning stage.

**3. Evaluation and comparison with baselines**

To my understanding, by default, RETRO-48B uses top documents for tasks like Open-domain QA (Implementation details). Then should a baseline that simply takes the same documents in input space (which is often called a retrieval-augmented generation) be used as a baseline? Also for me, Table 3 and 4 results look strange -- in prior work in retrieval-augmentation even on top of the SOTA LMs such as Codex, GPT-003, or Llama2-65B shows retrieval-augmentation gives large gain on tasks like open-domain QA. Why on RETRO-48B get even slightly better performance by removing encoders (and thus completely remove retrieved text)? Also even on Table 4 about long-form QA, the performance gap between w and w/o retrieval may not be statistically significant. While the authors claim this is evidence that pre-training with retrieval helps us to achieve a better decoder LM, I am rather confused about why their findings are different from prior work on retrieval augmentations. More quantitative and qualitative analysis on why this happens would help, but the paper doesn't have much ablations to address those questions.

Also, the gains in perplexity are impressive, there could be a certain risk of data leakage (e.g., a string similar to the test instance may be included in the database or training data), as also discussed in the RETRO paper. Probably more analysis and a sophisticated approach to mitigate the potential leak as in the RETRO paper could be helpful.

**Minor (but many) typos**

Although this is minor, there are at least 20 typos about pretraining (In the draft, many lines include the word "pertaining" -- I think the authors meant to say pretraining), including sub section titles. I recommend authors to fix this issue in the updated version of the draft.

**Questions:**

- How often the gate is set to zero during the instruction-tuning time?
- Do you have any insights into why removing encoders doesn't affect the performance?
- Have you tried different encoder moels rather than BERT-base? e.g., Contriever
- Did you do any analysis on the potential leak between train (database) /test splits in the pre-training corpus?

---

> ### Author Response · Authors · 2023-11-23
> **Thank you for your valuable comments**
>
> Thank you so much for the very detailed comments and feedback. They are really helpful in improving the quality of our paper. We also appreciate the reviewer liking our paper and pointing out the unclear descriptions.  We provide detailed clarifications below. We will improve our presentation per your suggestion in the updated draft.
>
> > 1. “Especially if the authors can open-source the code and checkpoints, it can inspire follow-up research on scaling retrieval-augmented LM pre-training.”
>   -  Many thanks for your comment. We will open-source the code and checkpoints. It is currently under institutional approval process.
>
> > 2. “I wish they provided ablations on different quantization methods to see whether different techniques can affect final performance or not, as conducting ablations on invidious quantization techniques can also be really expensive.”
>
> - Thank you for your valuable suggestions. Following your comments, we have included ablation studies on different quantization methods and hyper-parameters in Faiss index in Appendix Section B.3. Specifically, we include **two sets of ablation studies: on the Faiss training-time configurations and the query-time configurations**.
> - **For the ablation studies on the training-time configurations**, we follow the [official guide of Faiss](https://github.com/facebookresearch/faiss/wiki/Guidelines-to-choose-an-index) and initialize two Faiss indexes based on the sizes of two retrieval databases: the full pretraining corpus with 19B chunks and the Wikipedia corpus with 66M chunks.
> We applied product quantization to the full pretraining corpus to reduce the dimensionality and save the index memory to support loading the full pretraining corpus, while applying uncompressed flat encoding to the Wikipedia corpus as a comparison. We benchmark the querying speed for a batch of 40K dense embeddings and evaluate the query speed for two indexes.  The results is shown in the Table 1 below.
>
> *Table 1: Ablation studies on Faiss product quantization (PQ) on two different retrieval databases.*
>
> |                                           |                            |     Retrieval Index for Full   Pretraining Corpus    |     Retrieval Index for Wikipedia   Corpus    |
> |:-----------------------------------------:|:--------------------------:|:----------------------------------------------------:|:---------------------------------------------:|
> |                   #/ data                 |                            |                     19B                   |                   66M                  |
> |                Configuration              |     Dimension Reduction    |                       OPQ64_128                      |                  No Reduction                 |
> |                                           |     Approximate Search     |                   IVF4194304_HNSW32                  |                IVF262144_HNSW32               |
> |                                           |     Encoding               |                          PQ64                        |                  Flat Encoding                |
> |     Query Speed          (40k queries)    |             K=2            |                     0.004 s/query                    |                  0.01 s/query                 |
> |                                           |             K=20           |                     0.004 s/query                    |                  0.01 s/query                 |
> |                                           |            K=200           |                     0.0045 s/query                   |                  0.01 s/query                 |
> |                                           |            K=2000          |                     0.004 s/query                    |                  0.01 s/query                 |
> - From Table 1, we can see that applying product quantization can not only help compress the index and save memory usage but also help improve the query speed, which is critical when scaling up the retrieval database. We can also see that increasing the number of K for K nearest neighbor searchers barely impacts the query speed.

---

> ### Author Response · Authors · 2023-11-23
> **Thank you for your valuable comments**
>
> > 2. “I wish they provided ablations on different quantization methods to see whether different techniques can affect final performance or not, as conducting ablations on invidious quantization techniques can also be really expensive.”
> (continued)
> - **For the ablation studies on the query-time configurations**,  the hyper-parameter `nprobe` and `efSearch` play important roles in the query time of Faiss given our index configuration with interveted file index structures and HNSW graph. The detailed explanation of each parameter is shown in Table 2 below.
>
> *Table 2: Important querying-time hyper-parameters for our Faiss index*
>
> |     index type      |     Index class    |     runtime parameter    |     comments                                                       |
> |---------------------|--------------------|--------------------------|--------------------------------------------------------------------|
> |     IVF*, IMI2x*    |     IndexIVF*      |     nprobe               |     the main parameter to adjust the   speed-precision tradeoff    |
> |     HNSW*           |     IndexHNSW      |     efSearch             |     the depth of the HNSW search                                   |
>
> - To select a proper set of query-time hyper-parameters with a good tradeoff of recall and speed, we conduct ablation studies with varying `nprobe` and `efSearch`. Specifically, we use the retrieval index built on the whole pretraining corpus, query the index with randomly sampled 10K chunks from the pretraining corpus, and evaluate the recall accuracy of retrieving the query chunk itself given top-$K=2000$. The query time and corresponding recall accuracy with different hyper-parameters are shown in Table 3. We thus select `nprobe=4096` and `efSearch=32` in our retrieval-augmented pretraining, considering its fast querying time without sacrificing the recall accuracy significantly. We also added the ablation studies on Faiss index parameters in Appendix B.3.
>
> *Table 3: Trade-off between speed and recall accuracy given different query-time hyper-parameters.*
>
> |                                   |     Time (per query)    |     Acc (K=2000)    |
> |-----------------------------------|-------------------------|---------------------|
> |     efSearch=256, nprobe=65536    | 21ms                    |     1.00            |
> |     efSearch=64, nprobe=16384     |     6ms                 |     0.96            |
> |     efSearch=32,   nprobe=4096    | 4ms                     |     0.93            |
> |     efSearch=16, nprobe=4096      | 2.5ms                   |     0.88            |
>
>
> > 3. “Is retrieval or encoder indeed used instruction-tuning, or is skipped most of the time? Figure 1 seems to indicate that retrieval is skipped during instruction tuning while the text in Section 3.2. says retrieval is skipped only when retrieval context is not available. But there are no clear descriptions of what "retrieval context is not available". I listed detailed questions in Weaknesses and Questions. If it’s mostly a standard instruction tuning without retrieval on top of RETRO-48B, I don’t think it’s that technically novel.”
>
> - Sorry for the confusion. The retrieval and Retro encoder are NOT used during the instruction tuning stage, because none of the instruction tuning datasets comes with a pre-given retrieved context as you pointed out.  Indeed, there is nothing new in the instruction-tuning stage compared with standard instruction tuning for the GPT decoder. However, **the novel and interesting finding** is that the benefits of retrieval-augmented pretraining are unlocked after standard instruction tuning. We elaborate on the underlying reason through the ablation study in Section 5.3.
> - We elaborate on this novel finding in the following.  As shown in Figure 5 (a), for GPT and Retro above 8B, the improvement of zero-shot QA results is marginal w/o instruction tuning because these foundation models are bottlenecked by their poor instruction following abilities. After applying instructing tuning, the instruction following bottleneck has been removed. As shown in Figure 5 (b), the performances of both GPT-43B and Retro-48B are largely improved, so the benefit of retrieval-augmented pretraining becomes more pronounced. Specifically, InstructRetro excels at leveraging evidence to generate answers from retrieved context for downstream tasks, because it is pretrained to do generation given retrieved context during Retro-style pretraining.

---

> ### Author Response · Authors · 2023-11-23
> **Thank you for your valuable comments**
>
> > 4. “The experimental results indicate that removing the encoder block doesn’t affect the performance that much, which contradicts prior findings where retrieving documents gives large improvements on top of strong LLMs in tasks like OpenQA. While the authors argue this indicates pre-training with retrieval can enhance a decoder-only LM, this result makes me question the effectiveness of instruction tuning or the proposed model's ability to use retrieved context.”
>   - Sorry for the confusion. We want to clarify that removing the encoder block for downstream tasks doesn’t mean we don’t do retrieval for downstream tasks. Indeed, we are still performing retrieval-augmented generation (RAG) across all the experiments, and the retrieved contexts are placed on the decoder side for downstream tasks, which was found to be effective in improving Retro’s accuracy for QA in a previous study [1]. We have made it more clear in Section 5.1.
>
> [1] Wang et al. Shall We Pretrain Autoregressive Language Models with Retrieval? A Comprehensive Study. EMNLP 2023.
>
> > 5. “I suspect the limited deterioration may be from the limited performance of the retrieval component itself (i.e., the BERT encoder is far from the current SOTA retrieval system in the same parameter scale)”
>   - Thanks for the question. BERT-embedding based retriever is only used for pretraining. For downstream tasks, we either use the task-provided contexts (e.g., SQuAD 1.1 and 2.0), or use state-of-the-art retrievers to retrieve high-quality contexts from the task-specic corpus. For example, we follow the KILT [2] and use DPR retriever [3] for both Natural Question (NQ) and TriviaQA, and DRAGON+ retriever [4] for doc2dial and other long-form QA datasets. We have made it more clear in Section 4.1.
>
> [2] Petroni, Fabio et al. “KILT: a Benchmark for Knowledge Intensive Language Tasks.” North American Chapter of the Association for Computational Linguistics (2020).
>
> [3] Karpukhin, Vladimir et al. “Dense Passage Retrieval for Open-Domain Question Answering.” EMNLP (2020)
>
> [4] Lin, Sheng-Chieh et al. “How to Train Your DRAGON: Diverse Augmentation Towards Generalizable Dense Retrieval.” ArXiv abs/2302.07452 (2023)
>
> > 6. About “Technical novelty”
>
>   - Many thanks for your valuable suggestion about the pretraining part. We totally agree. We provide a more detailed analysis of individual quantization techniques in Appendix Section B, including the selection of hyper-parameters, ablation studies, and corresponding speedup. We hope this information will help the community to learn more about building a scalable retrieval index for retrieval-augmented pretraining at scale.
>   - Yes, unfreezing decoder parameters during continued pretraining (a.k.a. Retro-fitting) is a noticeable difference from the original Retro, and it turns out to be the key to obtaining a better decoder-only LLM after standard instruction tuning. Otherwise, Retro’s decoder will be the same as the original GPT before and after instruction tuning, because our instruction tuning is standard for decoder-only GPT without using retrieval.
>   - Contribution about instruction tuning: As we elaborate in question 2 and Section 5.3 in paper, the novelty is not about the method but the interesting finding.  We find that the benefits of retrieval-augmented pretraining become much more pronounced (see Figure 4 (b)) after standard instruction tuning by removing the bottleneck on instruction following ability.
>
> > 7. “The description says that the gate is set to zero when retrieved context is not available, but it's unclear what retrieved-context is unavailable means. None of the instruction-following datasets such as OpenAssistant comes with a pre-given retrieved document. Does this mean retrieval is always off during pre-training? Or it is used for some datasets with pre-given context? The lack of details makes it difficult what's new for the instruction-tuning stage. ”
>
> - As we elaborate for question 3, the gate is always on during retrieval-augmented pretraining and always set to zero during instruction tuning and downstream task evaluation. We also add a Figure 4 to better elaborate the gate mechanism.

---

> ### Author Response · Authors · 2023-11-23
> **Thank you for your valuable comments**
>
> > 8. “Then should a baseline that simply takes the same documents in input space (which is often called a retrieval-augmented generation) be used as a baseline. The performance gap between w and w/o retrieval may not be statistically significant. While the authors claim this is evidence that pre-training with retrieval helps us to achieve a better decoder LM, I am rather confused about why their findings are different from prior work on retrieval augmentation. In prior work retrieval-augmentation gives large gain on tasks like open-domain QA. More quantitative and qualitative analysis on why this happens would help, but the paper doesn't have much ablations to address those questions.”
>
> - Thanks for pointing it out. Note that, the GPT-Instruct is exactly the retrieval-augmented generation (RAG) model, which takes the same retrieved context for its input as IntructRetro on downstream tasks. For clarity, we rename GPT-Instruct as $\text{GPT}_\text{RAG}$-Instruct across the updated draft.
> - We also follow your suggestion and include an ablation study on the retrieval augmentation to demonstrate the accuracy gap between w/ and w/o retrieval on open-domain QA tasks NQ and TriviaQA. The results are shown in Table 4 and Table 5.
>
> *Table 4: Accuracy gap between w/ and w/o RAG on NQ*
>
> |   |                                  | GPT-Instruct | InstructRetro |
> |---|----------------------------------|--------------|---------------|
> |   | w/o retrieval augmentation (RAG) | 21.2         | **21.8**          |
> |   | w/ retrieval augmentation (RAG)  | 37.0         | **38.9**          |
>
>
>
> *Table 5: Accuracy gap between w/ and w/o RAG on TriviaQA*
>
> |   |                                  | GPT-Instruct | InstructRetro |
> |---|----------------------------------|--------------|---------------|
> |   | w/o retrieval augmentation (RAG) | 53.6         | **54.5**          |
> |   | w/ retrieval augmentation (RAG)  | 65.0         | **65.6**          |
>
>
> - From Table 4 and Table 5, we can see that the accuracy gap between with or without RAG is significant, which aligns with the prior work on retrieval augmentation. For example, the EM scores of NQ surges from 21.8 to 38.9 for InstructRetro. The improvement margin is significant GPT-Instruct as well.
> Moreover, we observe that InstructRetro consistently outperforms GPT-Instruct across different tasks, even without retrieval augmentation.
> This further confirms that instruction tuning after retrieval-augmented pretraining can help yield a better GPT decoder. We also added the ablation studies in Appendix D.2.
>
> >  9. “Why on RETRO-48B get even slightly better performance by removing encoders (and thus completely removing retrieved text)? Do you have any insights into why removing encoders doesn't affect the performance?”
> - Thanks for raising the question. Removing the encoder for downstream tasks doesn’t mean we don’t do retrieval for downstream tasks. Indeed, the retrieved texts are placed on the decoder side of Retro for downstream tasks, which was found to be effective in improving Retro’s performance for open-domain QA in a previous study [1].
> - In this way, the Retro’s decoder can be employed in the same way as the GPT decoder for retrieval-augmented generation. Moreover, it has much better accuracies than GPT i.e., see comparison between InstructRetro vs. $\text{GPT}_\text{RAG}$-Instruct.
> - To understand why removing encoders instead doesn’t affect the performance, we attribute this for two reasons: i) The retrieval enhanced decoder of Retro could already well unitize the retrieved context, so appending the retrieved context at the encoder side makes marginal difference. ii) As we clarified before, the encoder of Retro has not been optimized during instruction tuning. We hyposize a retrieval-augmented instruction tuning, that also optimizes the encoder, can make the difference. We leave it as future work.
>
>
> > 10. “Minor (but many) typos”
> - Many thanks for raising the issue. We have fixed the typos in the updated version of the draft.
>
> > 11. “How often the gate is set to zero during the instruction-tuning time?”
> - As we elaborate for question 3 and question 7, the gate is always on during retrieval-augmented pretraining and always set to zero during instruction tuning and downstream task evaluation. We also add Figure 4 to better elaborate the gate mechanism.

---

> ### Author Response · Authors · 2023-11-23
> **Thank you for your valuable comments**
>
> > 12. “Have you tried different encoder models rather than BERT-base? e.g., Contriever”
> - Thanks for the interesting question. We haven’t tried using contriever in retrieval-augmented pretraining yet. It is an interesting direction for future study. However, we would like to note that (1) we are using BERT-large instead of BERT-base to embed the retrieval text corpus; (2) we are using state-of-the-art retrievers to retrieve high-quality contexts from the task-specic corpus. For example, we follow the KILT [2] and use DPR retriever [3] for both Natural Question (NQ) and TriviaQA, and DRAGON+ retriever [4] for doc2dial and other long-form QA datasets. We have made it more clear in Section 5.1.
>
>
> > 13. “Did you do any analysis on the potential leak between train (database) / test splits in the pre-training corpus?”
> - Thank you for the valuable comments. We would like to clarify that our pretraining corpus has gone through multi-rounds of filtering and deduplications to minimize the potential data leakage. Specifically, our data curation strategy includes document-level deduplication, quality filtering, and downstream task deduplication to curate our final pretraining corpus and retrierval database. For deduplication, we follow the strategies from C4 [5], MT-NLG [6], and Gopher [7] and designed different 23 rules to deduplicate different documents, including thresholding the repeating n-gram fraction, N-gram LM Perplexity, etc. We believe that the data curation steps can make sure the whole corpus is well deduplicated and minize the potential risks between train and test splits.
> - We also note that the performance improvement of InstructRetro from GPT-instruct is not from the data leakage, because these two models use the same amount of training data in the following three stages. Stage I) Both of them are built on the same pretrained GPT-43B on 1.1T tokens. Stage II) Then, both of them are continually pretrained on 100B tokens. In this step, Retro 48 is doing retrieval on a 1.2T token database, which is simply the combination of above 1.1T and 100B pretraining corpus. Stage III) Both models are fine-tuned on the same instruction tuning set. As a result, data leakage during training could not be the reason for the performance improvement of InstrctRetro. It is also worth mentioning that InstructRetro didn’t use the 1.2T retrieval database for downstream tasks at inference time.
>
> [5] Raffel, Colin et al. “Exploring the Limits of Transfer Learning with a Unified Text-to-Text Transformer.” J. Mach. Learn. Res. 21 (2019): 140:1-140:67.
>
> [6] Smith, S., Patwary, M., Norick, B., LeGresley, P., Rajbhandari, S., Casper, J., Liu, Z., Prabhumoye, S., Zerveas, G., Korthikanti, V.A., Zhang, E., Child, R., Aminabadi, R.Y., Bernauer, J., Song, X., Shoeybi, M., He, Y., Houston, M., Tiwary, S., & Catanzaro, B. (2022). Using DeepSpeed and Megatron to Train Megatron-Turing NLG 530B, A Large-Scale Generative Language Model. ArXiv, abs/2201.11990.
>
> [7] Rae, Jack W. et al. “Scaling Language Models: Methods, Analysis & Insights from Training Gopher.” ArXiv abs/2112.11446 (2021)

---

> > ### Comment · Reviewer_gdXh · 2023-11-30
> > **Thank you so much for all of the detailed response!**
> >
> > Dear authors,
> >
> > Thank you so much for providing all of the detailed response and adding new ablations and experiments. I am really appreciate all of the modifications that the authors made within short amount of time, and I think my concerns is mostly addressed. I increase my score.

---

### Official Review · Reviewer_A1jh · 2023-11-02

**Soundness:** 3 good
**Presentation:** 3 good
**Contribution:** 3 good
**Rating:** 6
**Confidence:** 3

**Summary:**

The paper presents "Retro 48B", a large language model pretrained with retrieval before instruction tuning. Compared to the 43B GPT, Retro 48B showcases enhanced perplexity performance. After instruction tuning, InstructRetro significantly outperforms in zero-shot question answering tasks. Notably, removing InstructRetro's encoder and using only its decoder yields similar results. This highlights the decoder's enhanced capability for QA when pretrained with retrieval.

**Strengths:**

- The research highlights the benefits of continuing pretraining with retrieval mechanisms before proceeding to instruction tuning, a methodology that hasn't been extensively explored before.
- The paper brings to light the enhanced capability of the decoder in context incorporation for QA tasks when it's pretrained with retrieval, offering a fresh perspective on the potential of decoders in LLMs.
- The empirical results look nice. Retro 48B demonstrates enhanced perplexity performance when compared to the established 43B GPT model.

**Weaknesses:**

- The scalability, computational costs, and efficiency of training such models might be a concern.
- A more diverse set of metrics, especially some human evaluation, could provide a comprehensive understanding of the model's performance.

**Questions:**

- What is the computational overhead for introducing the retrieval part?
- How did the authors balance the blend of instruction tuning datasets?
- How does the authors explain the difference of question acc only appears after instruction finetuning.

---

> ### Author Response · Authors · 2023-11-23
> **Thank you for your valuable comments**
>
> We thank the reviewer for the valuable comments and feedback. We appreciate that the reviewer finds our work brings to light the enhanced capability of the decoder in context incorporation for QA tasks when it's pretrained with retrieval and our empirical results nice. We will address your comment in the following.
>
> > 1. “The scalability, computational costs, and efficiency of training such models might be a concern … What is the computational overhead for introducing the retrieval part?”
>   - Thank you for the valuable comments. We will elaborate on the additional computational cost for Retro-48B. It consists of two parts:
>     - i) **Retrieval index building cost**. Specifically, it consists of the following major steps:
>       - **Embedding the retrieval database into dense BERT embeddings.** Given the chunk size of $m=64$ tokens, we embed every chunk of text corpus with BERT-large-cased. The computational cost to embed the text corpus is around 6.22M chunks per GPU hour given one A100 GPU. For our 19B chunk database, it takes around 3054 GPU hours in total.
>       - **Train the Faiss index.** This involves determining a smaller number of centroids to cluster the whole corpus embeddings and initializing the HNSW graph. The computational cost of training the Faiss index depends on the number of corpus embeddings and the number of centroids. Given our setup, we train the faiss index based on 600M chunks uniformly sampled from the whole retrieval database. The computational cost of this step is less than 4 hours with one DGX A100 node.
>       - **Add the embedded corpus to the Faiss index.** After the index has been trained, the index centroids and HNSW graph are determined, but the index itself is still empty. In this step, we add the whole dense corpus embeddings to the index data structure. The computational cost of adding the corpus to the index is around 192 CPU hours within one DGX A100 node. Moreover, it can be purely done within a CPU node to save computational cost.
>       - **Query the Faiss index.** As mentioned above, we can achieve *4ms* per query over the whole pretraining corpus via batch queries averaged for each chunk with less than 1TB memory usage as our max throughput. The computational cost to query over 100B tokens in our continued pretraining step is around 1736 CPU hours within a DGX A100 node. Moreover, this step can also be purely done within a CPU node to save computational cost and can run in parallel to further speed up the querying.
>       - In summary, **the overall computational cost of building Faiss index is marginal compared to the pretraining cost**, especially considering the benefits of retrieval-augmentation pretraining, which further unlocks the potential of instruction tuning. Thus we believe that it is a promising direction to pretrain with retrieval augmentation.
>      - ii) **Additional pretraining cost**. We have provided our computation costs associated with GPT and Retro of different sizes below for continued pretraining on 100B tokens. All of our experiments are done on the DGX-A100 node with 8x A100 GPUs.
>
> |      | GPT on 100B token | Retro on 100B token | Additional Overhead |
> |------|-------------------|---------------------|---------------------|
> | 800M | 1408 GPU Hours    | 1920 GPU Hours      | 36%                 |
> | 2b   | 3226 GPU Hours    | 4096 GPU Hours      | 27%                 |
> | 8b   | 12698 GPU Hours   | 17325 GPU Hours     | 37%                 |
> | 22b  | 37888 GPU Hours   | 52152 GPU Hours     | 37%                 |
> | 43b  | 53329 GPU Hours   | 69995 GPU Hours     | 31%                 |
>
>   - From the table, we can see that the overhead involved in pretraining Retro is around 35% on average. Considering consistent improvements in downstream task accuracy and its capability to retrieve from trillions of tokens, we believe pretraining Retro is a promising, practical, and scalable direction for future LLMs. We have included more details of computational cost in Appendix A.3 and Appendix B.2 in the revision.

---

> ### Author Response · Authors · 2023-11-23
> **Thank you for your valuable comments**
>
> > 2. “A more diverse set of metrics, especially some human evaluation, could provide a  comprehensive understanding of the model's performance.”
>   - Thank you for your valuable comments. We follow your suggestions and conduct another round of human evaluation on long-form QA tasks to further examine the quality of the generated long answers. Specifically, we uniformly sample 600 examples containing prompts, ground truth answers, and responses from InstructRetro and GPT-Instruct in total, with 200 sampled from doc2dial, 200 from car manual doc #1, and 200 from car manual doc #2. We ask the annotators to compare the answer quality and choose from three options: (1) Model 1 Response, (2) Model 2 Response, (3) Equivalent. Note that responses from InstructRetro and GPT-Instruct are randomly shuffled for model 1 or model 2. Each sample is annotated with 5 different annotators. We report the percentage of human annotations in the table below.
>
> |                   | GPT-Instruct | InstructRetro | Equivalent |
> |-------------------|--------------|---------------|------------|
> | doc2dial          | 19%          | **33%**           | 48%        |
> | car manual doc #1 | 27%          | **31%**           | 42%        |
> | car manual doc #2 | 23%          | **25%**           | 52%        |
>
> - From the table above, we can see that in around half of the cases both models are giving reasonable and satisfactory results. In the rest of the cases, human annotators prefer the responses from InstructRetro more than GPT-Instruct consistently across three datasets. This further confirms that InstructRetro can generate more relevant answers than GPT-Instruct in long-form QA tasks.
>
> > 3. “How did the authors balance the blend of instruction tuning datasets?”
>   - Thank you for the question. While the blending ratio of instruction tuning datasets can be heuristic, there are several rules of thumb: (1) When deciding the blending ratio, we look at the number of total samples of each dataset. A dataset with more samples is given a higher ratio, but we also apply a cap to the max ratio of each dataset. (2) When generating a blend, we look at the associated number of epochs per dataset and make sure it looks reasonable. For example, no dataset should be trained for a lot of epochs to avoid overfitting. (3) We also adjust the blending ratio according to the dataset quality through manual inspection. (4) We have tried more than 10 different blending ratios and run small-scale experiments on each blend, until we finalized the best blending ratio. We leave it as an interesting future direction to construct better instruction tuning datasets for retrieval-augmented LMs.
>
>
> > 4. “How does the authors explain the difference of question acc only appears after instruction finetuning?”
>   - Thank you for the insightful question.  For GPT and Retro above 8B, the improvement of zero-shot results are marginal without instruction tuning as shown in Figure 5 (a), because these foundation models are bottlenecked by their poor instruction-following abilities. After instructing tuning, the instruction following bottleneck has been removed (as shown in Figure 5 (b), the performances of both GPT and Retro are largely improved), so the benefit of Retro-style pretraining becomes more pronounced. Specifically, InstructRetro excels at leveraging evidence to generate answers from retrieved context for downstream tasks, because it is trained to do generation given retrieved context during Retro-style pretraining.

---

### Official Review · Reviewer_p68m · 2023-11-08

**Soundness:** 3 good
**Presentation:** 3 good
**Contribution:** 3 good
**Rating:** 5
**Confidence:** 3

**Summary:**

The paper presents a significant advancement in large language models (LLMs) by leveraging retrieval-based pretraining. The authors introduce Retro 48B, the largest LLM pretrained with retrieval prior to instruction tuning, addressing limitations posed by the size of existing models. By augmenting the 43B GPT model with an additional 100 billion tokens retrieved from a vast database of 1.2 trillion tokens, they achieve a notable improvement in perplexity. Following instruction tuning, InstructRetro 48B demonstrates substantial enhancements in zero-shot question-answering tasks, surpassing its GPT counterpart. Overall, this work highlights the potential for further advancements in LLMs through retrieval-based training.

**Strengths:**

1. Proposal of the largest LLM pretrained with retrieval.
2. Good zero-shot question-answering capability.

**Weaknesses:**

1. The model is only evaluated on QA tasks
2. The paper should better include the results of retrieval-augmented LMs.
3. The paper could benefit from providing additional explanations or motivation regarding how retrieval-augmented training enhances the performance of LLMs. Could this improvement be attributed to potential data leakage during the training of Retro 48B or continued training with more data?

**Questions:**

n/a

---

> ### Author Response · Authors · 2023-11-23
> **Thank you for your valuable comments**
>
> We thank the reviewer for the valuable comments and feedback. We appreciate that the reviewer finds our paper demonstrates significant empirical results. We will address your concerns in the following.
>
> > 1. “The model is only evaluated on QA tasks.”
>   - Thank you for your suggestion. Following your suggestion, we’ve conducted experiments on three summarization tasks: QMSum [1], SummScreenFD [2], and GovReport [3]. Following the official metrics, we report the geometric mean of ROUGE scores (i.e., ROUGE1/2/L) for these summarization tasks. The zero-shot evaluation results are shown in the table below.
>
> | ROUGE scores  | GovReport | SummScreenFD | QMSum |
> |---------------|-----------|--------------|-------|
> | GPT-Instruct  |     12.59 |        10.43 | 15.06 |
> | InstructRetro |     **17.46** |        **10.93** | **15.61** |
>
> From the table above, we observe that **InstructRetro consistently outperforms the GPT-Instruct on these summarization tasks**, especially on the GovReport dataset with 4.87 ROUGE score improvement. This experiment further confirms the generalizability of IntructRetro after instruction tuning and indicates that Instruction tuning post retrieval-augmented pretraining yields a better GPT decoder. We also add the summarization results in Appendix D.3.
>
> [1] Zhong, Ming, et al. "QMSum: A new benchmark for query-based multi-domain meeting summarization." arXiv preprint arXiv:2104.05938 (2021).
>
> [2] Chen, Mingda et al. “SummScreen: A Dataset for Abstractive Screenplay Summarization.” ArXiv abs/2104.07091 (2021)
>
> [3] Huang, Luyang Robby et al. “Efficient Attentions for Long Document Summarization.” ArXiv abs/2104.02112 (2021)
>
> > 2. “The paper should better include the results of retrieval-augmented LMs.”
>   - Thank you for pointing it out. We would like to clarify that we **indeed provide the results of retrieval-augmented LMs, including Retro 7.5B, Retro++9B, Atlas 11B, and Raven 11B in Table 1**. Importantly, the GPT-Instruct 43B baseline is retrieval-augmented as well. Both GPT-Instruct 43B and InstructRetro 48B take the same retrieved context as model input at inference time. For clarity, we rename GPT-Instruct as $\text{GPT}_\text{RAG}$-Instruct across the updated draft. We have revised Table 1 in Section 5 to make the retrieval-augmented LM baselines more clear.
>
>
> > 3. “The paper could benefit from providing additional explanations or motivation regarding how retrieval-augmented training enhances the performance of LLMs. Could this improvement be attributed to potential data leakage during the training of Retro 48B or continued training with more data?.”
>   - Thanks for raising the question. **The performance improvement of InstructRetro from GPT-instruct is not from the data leakage, because these two models use the same amount of training data in the following three stages**. Stage I) Both of them are built on the same pretrained GPT-43B on 1.1T tokens. Stage II) Then, both of them are continually pretrained on 100B tokens. In this step, Retro 48 is doing retrieval on a 1.2T token database, which is simply the combination of above 1.1T and 100B pretraining corpus. Stage III) Both models are fine-tuned on the same instruction tuning set. As a result, data leakage during training could not be the reason for the performance improvement of InstrctRetro. It is also worth mentioning that InstructRetro didn’t use the 1.2T retrieval database for downstream tasks at inference time.
>   - In section 5.3 of our paper, we provide a detailed ablation study to analyze the source of the improvement for InstructRetro. Specifically, the retrieval-augmented pre-training of Retro 48B enhances the capability of leveraging relevant evidence from retrieved or provided context at test time, because the model is continually pre-trained to do next-token-prediction by feeding the retrieved context from the database. The improvement of zero-shot results are marginal without instruction tuning as shown in Figure 5 (a), because the model is bottlenecked by its poor instruction following capability. The improvement becomes significant after instructing tuning as shown in Figure 5 (b).

---

### Author Response · Authors · 2023-11-23
**General Updates**

We thank all the reviewers for their comments and valuable feedback. We have made the following major updates following the reviews to further improve our work.

1. Following the suggestions from Reviewer p68m, we add more experimental results on three summarization tasks in Appendix D.3.

2. Following the suggestions from Reviewer p68m and Reviewer gdXh, we revised Table 1 in Section 5 to make the retrieval-augmented LM baselines more clear. We also rename GPT-Instruct as $\text{GPT}_\text{RAG}$-Instruct across the updated draft.

3. Following the suggestion from Reviewer A1jh, we add detailed computational cost for pretraining Retro, including retrieval index building cost and continued pretraining cost in Appendix B.2 and Appendix A.3.

4. Following the suggestion from Reviewer A1jh, we add human evaluation on long-form QA tasks to further examine the quality of the generated long answers.

5. Following the suggestions from Reviewer gdXh, we provide detailed ablation studies on different quantization methods and hyper-parameters in Faiss index in Appendix Section B.3. Specifically, we include two sets of ablation studies: on the Faiss training-time configurations and the query-time configurations.

6. Following the suggestions from Reviewer gdXh, we include an ablation study on the retrieval augmentation to demonstrate the accuracy gap between w/ and w/o retrieval on open-domain QA tasks NQ and TriviaQA in Appendix D.2.

7. Following the suggestions from Reviewer gdXh, we have fixed the typos and further polished the writing of the draft.


All updates are highlighted in blue in our revision. If the manuscript is accepted, all contents in blue in the main text will remain for the camera-ready version.

---

### Meta-Review · Area_Chair_6Sun · 2023-12-06

**Metareview:**

The paper introduces Retro 48B, a large language model (LLM) pretrained with retrieval before instruction tuning. The model is built by augmenting the 43B GPT model with an additional 100 billion tokens retrieved from a database of 1.2 trillion tokens. The model shows significant improvement in perplexity and zero-shot question answering tasks. The authors also find that the encoder can be removed from the architecture, with the decoder alone achieving comparable results.

The reviewers appreciate the advancement in LLMs presented in the paper. They acknowledge the potential of retrieval-based pretraining and the promising results shown by Retro 48B. However, they also raise concerns about the paper's focus on QA tasks and the lack of diverse metrics. They also question the scalability and computational costs of training such large models. Main concern of this work is its evaluation, which focuses on QA. Although it was a major task of RAG models, but it is different from the diverse applications of LLMs. (The author responded with summarization results, but the improvement on SummScreenFD and QMSum seems marginal, and more ablations are also needed here.)

**Justification For Why Not Higher Score:**

Firstly, the paper's evaluation focuses primarily on question-answering tasks, limiting the understanding of the model's performance across a diverse range of applications. Secondly, the reviewers raised questions about the technical novelty of the work and the clarity of the instruction-tuning process. These issues need to be addressed to fully appreciate the contributions of the paper.

**Justification For Why Not Lower Score:**

N/A

---

### Decision · Program_Chairs · 2024-01-16

Reject